# The physical and psychological well-being after a pulmonary embolism across age and comorbidities – Evidence from focus group interviews

Caroline Sindet-Pedersen[1]*, Nina Nouhravesh[1], Simone Hofman Rosenkranz[1], Sophie Fredslund Madsen[1], Morten Lamberts[1,4] Thomas Kümler[1,5], Gunnar Gislason[1,2], Nina Føns Johnsen[2], Anette Arbjerg Højen[3]

1 Department of Cardiology, Copenhagen University Hospital, Herlev and Gentofte Hospital, Herlev, Denmark, 2 The Danish Heart Foundation, 1127 Copenhagen K, Copenhagen, Denmark, 3 Danish Center for Health Services Research, Department of Clinical Medicine, Aalborg University and Aalborg University Hospital, Aalborg Denmark, 4 Complications Research, Steno Diabetes Center, Copenhagen, 5 Department of Clinical Medicine, University of Copenhagen, Denmark

* carolinesindet@gmail.com

## Abstract

### Background

Few studies have investigated the psychological and physical consequences of having experienced a pulmonary embolism (PE), and most patients with PE are not offered proper follow-up.

### Objective

To gain qualitative insight into the physical and psychological well-being across age and comorbidities and to investigate the patients need for rehabilitative strategies in patients with PE.

### Methods

Qualitative data was obtained through focus group interviews. Patients were recruited from an outpatient thrombosis clinic at Herlev and Gentofte hospital in Denmark, based on the principles of maximum variation strategy which included age, concomitant disease, risk factors for pulmonary embolism (pregnancy, infection, cancer, and recurrence). Data was analysed using inductive thematic analysis.

### Results

Six focus group interviews were conducted with a total of 17 participants being interviewed. Results showed that a significant degree of symptoms was experienced after PE. The emotional reactions experienced was largely affected by disease experience, and varied according to age, cancer status and PE in relation to pregnancy.

**Data availability statement:** The data underlying this article cannot be Due to legal and ethical obligations under the EU General Data Protection Regulation (GDPR) and national data protection legislation, we are not permitted to share full raw data, even in anonymized form, as re-identification remains a risk, particularly given the small sample size and context-specific narratives. Participants were assured during the informed consent process that their full transcripts would not be shared outside the research team. Data may be shared with datas, upon reasonable request from forskningsjura. rigshospitalet@regionh.dk.

**Funding:** This study has been funded by the Danish Heart Foundation, Grant number: A9530. And by an unrestricted grant from Læge Sofus Carl Emil Friis og Hustru Olga Doris Friis' Legat. The funders had no role in study design, data collection and analysis, decision to publish, or preparation of the manuscript.

**Competing interests:** CS: None, NN: Speaker fees from Astra Zeneca and Bayer, AN: Payment or honoraria for lectures, presentations, speakers bureaus, manuscript writing or educational events from Bayer, MSD, Pfizer, Leo Pharma and BMS, SF: none, TK: MSD, lecture on cardiology, Support attending conferences from Medtronic, NF: none, SM: none, GG: None, ML: payment for honoraria for lectures, presentations, speakers bureaus, manuscript writing or educational events from BMS, Pfizer, Bayer, Astra Zeneca, advisory board, Pfizer and Astra Zeneca.

Anticoagulation therapy was perceived as a life saver, yet it also contributed to a perception of being chronically ill, illustrating a medication conundrum. Lastly, confusion and frustration regarding follow-up care was prominent with a perception of limited guidance and limited information about potential rehabilitation strategies following PE.

## Conclusion

This study showed that the patients' experiences, worries and needs are different according to age and comorbidities, which indicates that interventions aimed at improving outcomes for these patients should be targeted accordingly.

## Introduction

Pulmonary embolism (PE), defined as blood clots that travel to and obstruct the pulmonary arteries, is the most severe presentation of venous thromboembolism [1]. The group of patients with PE is heterogenous, ranging from young women who developed the PE in relation to oral contraceptive use or pregnancy to cancer patients and patients with no underlying disease or risk factors. [2,3]

Previous studies have shown that up to 50% of patients with PE experience long-term functional limitations such as dyspnea, decreased physical capacity and depression [3,4]. While these studies describe the prevalence and severity of long-term complications, they provide limited insight into how patients experience, interpret, and manage these physical and psychological consequences of PE in everyday life. Some qualitative studies have explored the patients' lived experiences, [5–9]. However, Limitations of previous studies include small sample sizes and the inclusion of selected patient populations; for example, one study focused on highly comorbid patients, while another recruited participants from a randomized trial of physical exercise after PE, which may have introduced selection bias, as healthier and more resourceful patients are more likely to participate in such studies. [6–9] Thus, there is a need for a study including a broader and more heterogeneous patient population, using qualitative methods to achieve a more comprehensive understanding of the physical and psychological consequences of PE. Additionally, there is an unmet need for studies investigating patients' perspectives on follow-up care and rehabilitation strategies after PE. Thus, the **aim** of this study is to gain insight into the physical and psychological well-being across age and comorbidities and to gain insight into the patients need for rehabilitative strategies among patients with PE.

### Methods

Qualitative data was obtained through focus group interviews. This methodology was chosen because it allows for the multiplicity of views to emerge, as opposed to individual interviews where it is the individual participants feelings and attitudes of individual participants are obtained [10,11]. In addition, it prompts a snowballing effect where participants responses may trigger other participants responses, whereby

the richness of data is increased. [10,11] The focus group interviews followed an interview guide (supplementary table 1), which included four main themes; psychosocial well-being, physical well-being, anticoagulation therapy and need for rehabilitation. The questions were open ended to make the participant reflect on their experiences and perspectives. The SRQR (Standard for reporting qualitative research) guideline for reporting qualitative research was followed during the design of the study and reporting of the results [12].

## Study setting and participants

Study participants were recruited between December 1, 2021, and March 31, 2022, at a large teaching hospital in Denmark, from the outpatient thrombosis clinic by two medical doctors (ML and TK).After recruitment, participants were contacted by CS (female, first author and postdoctoral researcher), who conducted the interviews with assistance from SM and NN, who took notes. In the beginning of all interviews CS gave an introduction to herself and explained the participants about the study. All participants were asked to fill out an information form at the end of the interview, which included basic questions about their age, sex, educational level, comorbidities, and risk factors for their PE.

Participants were recruited using a purposeful maximum variation sampling strategy to capture a broad range of experiences with PE. Sampling criteria included age, sex, comorbidities, and PE-related risk factors (including pregnancy, infection, cancer, thrombophilia, and recurrence). To facilitate meaningful interaction and in-depth discussion, focus groups were strategically composed based on shared characteristics, such as age group, pregnancy-related PE, or cancer status, thereby ensuring a common frame of reference beyond having experienced PE. This approach allowed exploration of both shared and contrasting experiences across patient groups.The number of focus groups and patients was guided by the principle of meaning saturation, that is the point at which additional interviews no longer generated new insights relevant to the research questions. After each focus group, the research team discussed emerging findings and compared them with insights from previous interviews. Recruitment continued until two consecutive focus groups yielded no new relevant insights, indicating that saturation had been reached [13]. In total, six focus groups with 17 patients were conducted, which was sufficient to ensure both depth and variation in patient experiences. Participants were excluded if they were under the age of 18, did not speak Danish or were cognitively impaired.

The research team comprised clinicians, nurses, and health researchers with experience in thrombosis care and qualitative research. Researchers involved in data collection and analysis had prior clinical and academic experience with PE, which may have influenced the research process. Reflexivity was addressed through ongoing interdisciplinary discussions of emerging findings, consideration of alternative interpretations, and critical reflection on how researchers' professional backgrounds and preunderstandings could shape data collection and analysis.

## Data analyses

All interviews were audio-recorded and transcribed verbatim. Data were analysed using inductive thematic analysis, following the approach described by Braun and Clarke. [14,15] This method allows for the identification of patterned meanings within the data while remaining close to participants' expressed experiences. [14,15] The analysis proceeded through the following phases:: 1) Familiarization with data, which included reading and listening to the interviews, 2) Generation of initial codes done by primary investigator CS, 3) Aggregate codes in themes, 4) Go through themes and text, 4) Defining and naming of themes. By using this method, a coding system was first used to index the entire dataset, which is a constructive method for organising data [14,15]. Initial coding was conducted inductively and data-driven, allowing themes to emerge without the use of a predefined coding framework. Codes were primarily semantic, focusing on explicit meanings in the data [15]. Coding was undertaken by multiple members of the research team (CS, NM, SR, and AH). Analytic meetings were held regularly to discuss coding decisions, compare interpretations, and refine the developing codebook. Discrepancies were resolved through discussion until consensus was reached, thereby enhancing analytic rigor. Throughout the analytic process, new codes were added when relevant, allowing the researchers to remain open to unexpected

insights. Member checking was not conducted. Instead, credibility was enhanced through investigator triangulation, iterative team discussions, and careful grounding of themes in the data, The software QSR Nvivo10 was used for analysis and coding.

### Theoretical framework

To support interpretation of patients' experiences, this study draws on established theoretical frameworks. [16–18] Leventhal's Common-Sense Model of Illness Perceptions provides insight into how symptom experiences, fears of recurrence, and coping responses may shape life after PE. [16] Bury's concept of chronic illness as a biographical disruption offers a lens for understanding how unexpected illness can affect identity and everyday life, particularly among younger patients. [17] Finally, Wagner's Chronic Care Model informs the exploration of follow-up care and rehabilitation needs from a health systems perspective. [18]

### Ethics and consent

The study was approved by the data-responsible institution, the Capital Region of Denmark (approval number P-2021–684). The Danish Regional Committee on Health Research Ethics waived the requirement for formal ethical approval (reference number F-25012553), as the study did not involve biological material, clinical interventions, or changes to standard care, and qualitative interview studies are not considered biomedical research requiring ethical approval under Danish legislation.

All participants received oral and written information about the study aims and procedures. Oral informed consent was obtained at inclusion, followed by written informed consent prior to participation in the focus group interviews.Participants were informed that participation was voluntary and that they could withdraw their consent at any time without consequences. All data were handled in accordance with data protection regulations.

## Results

### Patient characteristics

A total of 29 patients were approached for an interview, resulting in a total of six focus group interviews with a total of 17 participants being interviewed. One focus group had only one participant due to cancellations from 1 participant on the interview day. The duration of the interviews ranged from 1 hour to 2 hours. The age of the participants ranged from 39 to 81. Two patients had cancer, and one participant had experienced a pulmonary embolism during pregnancy. All participants except one were taking oral anticoagulation medication at the time of the focus group interview. Participant characteristics are described in Table 1.

### Themes

**Significance of symptoms.** The significance of symptoms following PE was prominent in relation to the participants physical and psychological well-being across age and comorbidities. Table 2 describes the themes identified.

The participants experienced a varying degree of symptoms such as fatigue and dyspnea and for some, the symptoms persisted. This had a cascading impact as participants indicated heightened awareness of symptoms and their bodily sensations following their PE. For some participants resulting in them being overly cautious;*"[…] That's what I thought, I can't call an ambulance every time. When I think it's[life] all leaving me, C(54):"* Thus, the participants experienced hypervigilance and anxiety towards symptoms of a recurrent PE, and confusion regarding which symptoms to act on.

There was a clear difference in the significance of symptoms according to age. The younger participants felt that the PE had had a large impact on their everyday life, often expressing frustration and disappointment that their physical performance had not returned to the pre-PE level. For these participants, the persistence of symptoms clashed with their expectations of

**Table 1. Baseline characteristics.**

| informant | Sex | Age | Marital status | Edu-cation | Work status | Time since PE (months) | Pre-vious PE | Pre-vious DVT | Risk factors | Con-comittant disease | OAC* | Symptoms after PE |
|---|---|---|---|---|---|---|---|---|---|---|---|---|
| Informant 1 | M | 81 | Married | Bachelor | Retired | 9 | No | Yes | Infection | | Yes | Dyspnea |
| Informant 2 | M | 80 | Married | Bachelor | Retired | 4 | Yes | No | | Psoriasis | Yes | None |
| Informant 3 | M | 75 | Married | Vocational | Retired | 2 | Yes | No | | | Yes | None |
| Informant 4 | F | 60 | Co-habiting | Bachelor | Employed | 9 | No | No | | | Yes | Psychological |
| Informant 5 | M | 52 | Married | Bachelor | Employed | 4 | No | No | | | Yes | None |
| Informant 6 | F | 73 | Married | High school | Retired | 5 | No | No | Operation | Hyper-tension | Yes | Fattigue/Pressure chest, |
| Informant 7 | F | 79 | Widow | Bachelor | Retired | 6 | No | No | | Hypertension | Yes | Dyspnea/Pain and pressure in chest |
| Informant 8 | F | 74 | Widow | Vocational | Retired | 4 | No | No | Travel | | Yes | Dyspnea |
| Informant 9 | F | 65 | Married | Bachelor | Retired | 10 | No | No | Operation | | No | Dyspnea |
| Informant 10 | M | 44 | Married | Vocational | Employed | 19 | No | No | | Colitis Ulcerosa | Yes | Dyspnea/Pain and pressure in chest |
| Informant 11 | M | 42 | Married | Masters | Employed | 4 | No | Yes | | Factor 5 Leiden | Yes | None |
| Informant 12 | M | 51 | Married | Bachelor | Employed | 24 | No | Yes | | Factor 5 Leiden | Yes | Dyspnea |
| informant 13 | M | 63 | Divorced | Bachelor | Employed | 18 | Yes | No | | | Yes | Dyspnea/Fattigue |
| Informant 14 | F | 59 | Cohabit-ing | Short-Cycle higher | Transfer income | 6 | No | No | Infection/Cancer | Cancer | Yes | Dyspnea/Fattigue |
| Informant 15 | M | 66 | Married | Bachelor | Self employed | 6 | No | No | Cancer | Cancer | Yes | Dyspnea |
| Informant 16 | M | 69 | Married | Bachelor | Retired | 3 | No | No | Operation | Hypertension | Yes | None |
| Informant 17 | F | 39 | Married | Bachelor | Employed | 28 | No | No | Preg-nancy | Increased cardiolipin | Yes | Dyspnea/Leg pain |

*OAC: oral anticoagulation therapy,

health, vitality, and ability to engage fully in work, family, and leisure activities. In contrast, especially the elderly male partic-ipants were more likely to normalize their symptoms and attribute them to aging. The difference between younger and older participants was primarily related to expectations of physical performance. As one younger female participant described:

"…I do not have chest pain, but I feel like I have pain in my lungs actually and I cannot run 2 km. without stopping numerous times. I have been used to running 10 km Ma(38)."

Whereas an elderly participant explained:

"…I haven't felt disabled, but when you are 79, um, you cannot expect… you sort of have to accept that things are going the wrong way K(79)."

Thus, younger participants experienced their symptoms as a disruption to their anticipated trajectory of health and recovery, reinforcing a sense of being "seriously ill" at an unexpected stage of life. Older participants, however, tended to

**Table 2. Identified major themes, subthemes and sub-subthemes, quotes and implication/interpretation.**

| Main theme | Subtheme | Sub-subtheme | Representative quote(s) | Implications/ Interpretation |
|---|---|---|---|---|
| Significance of symptoms | Experienced symptoms | Increased awareness of symptoms | "[…] That's what I thought, I can't call an ambulance every time. When I think it's[life] all leaving me, (C, 54)." | Some participants experienced ongoing symptoms, which contributed to hypervigilance and anxiety about recurrence. |
| | | Symptoms according to age and expectations | "…I cannot run 2 km… I used to run 10 km (Ma, 38)."<br>"…when you are 79… you sort of have to accept it (K, 79)." | Younger participants expected full recovery, while older participants attributed symptoms to aging and were more accepting. |
| | Everyday life | Difficulty getting back to work | "It was insanely tiring… I worked 3–4 hours a day at max (M, 43)." | Fatigue and physical limitations made it difficult to return to work. |
| | | Illness intervening in daily life activities | "I am here now, and then one more thing today and then I can't go out tonight… It was not like that before, (A, 58)." | Symptoms significantly disrupted social participation and daily routines. |
| Emotional reaction | Disease experience | Not a big deal the second time | "The first time… but this time it wasn't a big deal really (M, 78)." | Prior experience of PE reduced fear, as participants felt more familiar with symptoms. |
| | | Young vs. old | "[…] it's terrible to get seriously ill when you're young, (M, 38)"<br>"So I am not sick... if Someone asks, then I say, well, I didn't feel anything before I was admitted to the hospital. I don't feel anything now. I can also move and stuff" (M(74) | Younger patients saw PE as disruptive and unexpected, intensifying the emotional burden, while older patients tended to normalize the illness and emphasize preserved function. This highlights age-related differences in expectations of health and recovery. |
| | | Cancer | "What worries me is the prostate cancer, where the blood clot is subordinate (K, 65)." | PE was seen as secondary to the primary cancer diagnosis. |
| | | Pregnancy | "It was a good thing it was my second child… I don't think I would have had more children (Ma, 38)." | Pregnancy-related PE heightened fear for both mother and child and influenced decisions about future pregnancies. |
| | Fear of what could have happened | Near-death experience | "…It is a near-death experience, or it could at least have been (P, 51)." | The perception of PE as life-threatening triggered existential reflection and anxiety. |
| | | Change in life perspective | "I think about whether I spend the time on what I want (P, 51)." | The experience prompted re-evaluation of life priorities. |
| | | Feeling alone | "I think it could have been a good idea if you had someone to talk to (Ma, 38)." | Participants expressed a need for psychological support and someone to share their experience with. |
| The conundrum anticoagulation therapy | Paradox of treatment | | "…I simply think it's like crutches, I can't do without them (C, 59)."<br>"…I feel more chronically ill, I have never felt ill before (K, 41)." | Patients experienced ambivalence: reassurance from treatment but also fear of bleeding and a chronic illness identity. |
| | Bleeding | | "Every time you cut your finger, it is almost impossible to stop… (H, 81)." | Side effects reinforced treatment as both necessary and burdensome. |
| Impact of insufficient information | Help and guidance after discharge | | "We are being left somewhat in the lurch… others get rehabilitation, we are just sent home (C, 54)." | Lack of structured follow-up left patients feeling abandoned compared to other disease groups. |
| | Rehabilitation through exercise | | yes, so if it is the case that rehabilitation could have improved the situation you ended up in by having a blood clot, then it would be the right thing to do, but you haven't really been told anything about that…(H, 81). | Lack of guidance on exercise left participants uncertain about its role in recovery, highlighting a gap in post-PE rehabilitation support. |
| | Control scan | | "I would like a scan that says the three blood clots are gone (A, 58)." | Absence of follow-up imaging caused uncertainty and undermined reassurance of treatment success. |

accept limitations as a natural part of aging and therefore described the impact as less disruptive. This contrast under-scores how age-related expectations shape illness perceptions: for younger patients, PE was a challenge to identity and future prospects, while for older patients it was more readily integrated into an existing narrative of declining health**.**

The significance of the symptoms experienced by the participants impacted the everyday lives of the participants. For some participants, physical exhaustion made it difficult to go back to work. *"I think it[work] was tough I got really tired. […]. It was insanely tiring. I think I worked 3 to 4 hours a day at a max.. (M (43))."* Furthermore, the significance of symptoms manifested in the experienced symptoms effect on social life activities. One participant described how she was only able to participate in a limited amount of daily activities and had to prioritize: *"I am here now, and then one more thing today and then I can't go out tonight. There is simply...there is simply no room for that….It was not like that before the blood clot in the lung, A (58)"*. Thus, the experienced symptoms affected the participants normal everyday life routine, with the PE symptoms significantly inhibiting daily activities.

### Emotional reaction

The emotional reactions experienced by the participants following the PE was largely related to thoughts of what could have happened, a change in life view and a sense of feeling lonely.

The emotional reactions experienced by the participants were found to vary according to PE disease experience, cancer status, pregnancy, and age. A participant with previous experience of PE, described the second PE as less fright-ful:: *"The first time it was really uncomfortable, because I felt that I could hardly stand up, but this time I felt that it wasn't a big deal really, M(78)"*. The emotional reaction following the PE was affected by the prior experience of the PE as one described when asked about being more aware of symptoms the second time *"I was not aware (red. of the symptoms) two years ago, but this time I was aware"* thus, they were familiar with the symptoms, had been living with the disease and knew what to expect. Likewise, the emotional reaction for the participants with cancer was affected by their cancer status. It was clear that the cancer was the predominant worry for these participants and not the PE:.*"[…]what worries me is the prostate cancer, where the blood clot in the lung is subordinate, K (65)"*

Having PE in relation to pregnancy also affected the emotional reaction. A participant described:

*"It was a good thing it was my second child, otherwise I don't think I would have had any more children... Even though they say nothing will happen. It's clear that you get a bit worried. I didn't have those worries the first time I was preg-nant. There has been a lot of illness in my pregnancy, Ma (38)".*

Thus, the combination of experiencing a life-changing experience (expecting a child) alongside having a potentially life-threatening disease (PE) caused worry and fear of engaging in a future pregnancy due to the risk of having another PE. Indeed, the experience of pregnancy-related PE enlarged the emotional reaction as the participant worried both for herself and her unborn child.

The emotional reaction following PE was influenced by the age of the participants. The young participants expressed that: *"[…] it's terrible to get seriously ill when you're young, (M38)"*, and described how they perceived PE as an "old mans disease". "*I had an expectation when I came here today that it would be another (older red.) generation sitting here, K (41):"* Thus, PE was somewhat unexpected among the younger participants, making them feel ill, while the elderly partic-ipants seemed less affected and more accepting. The younger participants experienced their PE as a disruption to their anticipated trajectory of health and recovery, reinforcing a sense of being "seriously ill" at an unexpected stage of life. Older participants, however, tended to accept limitations as a natural part of aging and therefore described the impact as less disruptive. This contrast underscores how age-related expectations shape illness perceptions: for younger patients, PE was a challenge to identity and future prospects, while for older patients it was more readily integrated into an existing narrative of declining health**.**

The worry about the counterfactual outcome of their experience was also pivotal in the participants emotional reaction following PE. A participant, expressed how the perception of the PE as a near-death-experience induced an emotional reaction:

*"Everyone has those thoughts (red. about what could have happened). It would be strange not to, considering how severe it is. I might be downplaying it, but it quite an intense experience….It is a near death experience, or it could at least have been, P (51)".*

For some participants, the reaction was delayed and came weeks after hospital discharge, and for some participants the emotional reaction materialized in changing the participants' life perspective: *"[…]so I would say that I think about, whether I spend the time on what I want, P (51)".* Thus, the perception of PE as a life-threatening event that could have had fatal consequences, made participants reevaluate their priorities in life.

Because of the emotional reaction the participants described feeling alone, and expressed a need for someone to talk to: "*I think it could have been a good idea if you had someone (to talk to), Ma (38)."* Thus, an opportunity to share thoughts and feelings was paramount. However, for some this was a dilemma as they did not want to burden their children and relatives.

### The conundrum of anticoagulation therapy

A third theme that emerged in relation to participants physical and psychological well-being across age and comorbidities was the conundrum of anticoagulation therapy. The participants described mixed emotions about the benefit/risk of anticoagulation therapy in terms of bleeding and VTE recurrence. On one side they described anticoagulation therapy as a life saver on the other side as something that could harm them. A female participant described anticoagulation therapy as her crutches: "*...He would much rather have me stop taking the blood thinners, and I told him that I simply think it's like crutches, I simply don't believe I can do without them, C (59)".* Whereas another participant expressed worry about side effects of the anticoagulation therapy; "*yes, it's a bit bad (taking the medication), because every time you cut your finger, it is almost impossible to stop (red. the bleeding) again. So I would like to have stopped if I could, but I can also see it from the doctor's point of view, if it is a second blood clot, then you can also get a third one, and it can be catastrophic. So I can see that. So I'll take them, H (81)".* Furthermore, the participants described the dilemma of being reassured that PE would not recure, while simultaneously feeling chronically ill as a result; "*.... It gives some reassurance in one way or another, but at the same time it is also... a change, I feel more chronically ill, I have never felt ill before, K (41)".* Thus, there was a reluctance to stop treatment, but at the same time a fear of side effects and a perception of being chronically ill when continuing anticoagulation therapy depicting a medication conundrum.

### Impact of insufficient information

The impact of insufficient information was prominent in relation to participants physical and psychological well-being across age and comorbidities. Participants expressed a great need for help and guidance after discharge and described to be confused about what they could expect in terms of follow-up care. A male participant described follow-up care was perceived a matter of cause: *"Only wonder on my part. Wonder. I myself am a craftsman and engineer and I would never, ever, after a project like that, not follow up on whether what I had done was in order…Never ever. And when it is human life we are talking about, it should be even more important"* Thus, it was clear that follow-up with appropriate information and guidance was critical for participants to feel that their experience was taken seriously and ultimate for their ability to return to everyday life. However, they lacked information and guidance on how to get their life back to normal and

questioned why they were not offered any form of follow-up, as a female participant described: *"[…] and what I really feel, is that we are being left somewhat in the lurch. For those who have a heart attack or critical illness, they receive a 16-week municipal rehabilitation course and we are just sent home, C(54)"*. The participants compared themselves to patients with other diseases and wondered why they were not offered similar levels of follow-up and rehabilitation, feeling as though they were just left to their own devices.

In addition, the participants expressed confusion as to why they were not offered a follow-up scan to confirm their PE had resolved. When one participant was asked how she would want to be helped further, she replied:*I would like a scan. I would like a scan that says the three blood clots are gone, A(58) "*. Thus, the control scan was by the participants perceived as a reassurance that the treatment for PE had been effective and that the PE would not recur. However, reviling an insufficient information and communication between the medical doctors and participants, in that it had not been properly explained why follow-up scans were not offered.

## Discussion

Our findings showed that a significant degree of symptoms was experienced after PE. The emotional reactions experienced was largely affected by the disease experience, varying according to age, cancer status and PE in relation to pregnancy. Anticoagulation therapy was perceived as a life saver, yet it also contributed to a perception of being chronically ill, illustrating a medication conundrum. Lastly, confusion and frustration regarding follow-up care was prominent with a perception of limited guidance and information about potential rehabilitation strategies following PE.

Previous studies have shown that patients experience a varying degree of symptoms following their PE, and that these symptoms can affect their everyday lives to a large degree [7,9]. This was confirmed by this study, with many participants experiencing persisting symptoms such as dyspnea and fattique, even years after their PE, to a degree that it affected their lives negatively. As has been described in other studies, many PE Participants experience hypervigilance after a PE, and this can be a sign of post-traumatic stress syndrome. [8,9,19] In this study, hypervigilance was observed for some of the participants, which escalated their fears of and negative beliefs about their health. Applying Leventhal's Common-Sense Model of Illness Perceptions provides further context for these findings. Persistent symptoms shaped participants' illness identity, while fears of recurrence reflected perceptions of severe consequences and an uncertain timeline. Hypervigilance further illustrates how bodily sensations became central to the illness identity and reinforced beliefs about unpredictable consequences, leading some participants to interpret minor symptoms as signs of recurrence. [16]

The impact of symptoms and emotional reaction was largely affected by the patient's disease experience. Our findings showed that the younger participants felt that the PE had had a greater impact on their lives compared with the older participants, reporting a higher degree of symptoms, anxiety and feelings that the anticoagulation therapy intruded in their lives and made them feel ill. This is very well in line with Burys theory on chronic illness as a biographical disruption, where it is described that younger patients may feel a biographical disruption when illness occurs ahead of time or unexpected, in which everyday life is disrupted by the illness [17]. As found in our study many of the younger participants, considered PE a disease of old age. In addition, the prospect of taking medication for the rest of their lives reminded them of their illness. [9,19] Thus, experiencing a PE at a younger age could be seen as biographical shift from the expected trajectory, thus something that should have occurred at a later point in life, if ever. [9,17] In addition, Faircloth argues that later in life, contrary to earlier in life, illness is experienced as a biographical flow, as the onset of illness and other health issues are to be expected with increasing age. [20,21] Further, because the disease affects fewer younger patients, they lack peers to talk to, unlike the older patients. This experience was especially present when PE was experienced during pregnancy. Meanwhile, the elderly generation possibly have more experience with disease or resources in the form of peers, who have gone through similar illness, which they can lean on and talk to, in order to help their recovery, [9,17] For the participants with cancer, it was clear that the cancer was the predominant worry, which has been thoroughly described by Nouhravesh et. al. [22]

                                                                                        

An interesting theme was the conundrum of taking anticoagulation therapy – on one hand, it was perceived as something positive, because the treatment protected them against a recurrent event, but on the other hand, there is a risk of bleeding associated with taking the medication. In addition, taking medication every day reminds them of the PE, making them feel chronically ill. This experience can be understood in light of Bury's concept of biographical disruption, where illness and its treatments intrude on everyday life and reinforce an altered sense of identity. [17] In the study of Rolving et.al, medication was described as a life saver and that adverse events were seen as something insignificant. [7] This was contrary to this study, where participants expressed worries about adverse events, but at the same time seeing the medication as something that helped them avoid a recurrent PE, which was something they feared. When discussing potentially ending treatment, Participants expressed concern about the outcome if this happened. This has also been described by Kirchberger et. al. [6]

Almost all participants described confusion about follow-up, the participants felt that they had not been offered sufficient follow-up or rehabilitation, despite the fact that a PE can be life threatening and possibly also as devasting as a myocardial infarction. It was clear that the participants were eager to undertake rehabilitation or other beneficial actions if advised to. Golemi et. al. identified a major theme about incomplete communication during transitional and follow-up care [23]. It was shown that inappropriate follow-up was a common cause of confusion and distress. It has been shown that key factors for successful management of PE with multidisciplinary teams and regular follow-ups, is associated with a decreased anxiety, and confusion among patients. [1] The lack of appropriate follow-up could possibly also have increased their anxiety around the PE. These findings align with Wagner's Chronic Care Model, which emphasizes proactive, structured, and multidisciplinary follow-up as central to chronic disease management. Applying this framework to PE survivors underscores how systematic, patient-centered care could reduce uncertainty, improve recovery, and address the current gaps in follow-up. [18] Applying Leventhal's Common-Sense Model of Illness Perceptions provides further context for these findings. Persistent symptoms shaped participants' illness identity, while fears of recurrence reflected perceptions of severe consequences and an uncertain timeline. Hypervigilance further illustrates how bodily sensations became central to the illness identity and reinforced beliefs about unpredictable consequences, leading some participants to interpret minor symptoms as signs of recurrence. [16] Similar to Danielsbacka et al. (2021), who highlighted that PE survivors need support to manage both psychological and physical symptoms and called for further research on optimal rehabilitation, [24] our study reinforces these needs but adds new insight by demonstrating age-related differences, the "medication conundrum," and the importance of tailoring follow-up to individual illness perceptions. our study reinforces these needs but adds new insight by demonstrating age-related differences, the "medication conundrum," and the importance of tailoring follow-up to individual illness perceptions..

Recent initiatives, such as the Danish Attend-PE model, illustrate how these gaps might be addressed through structured follow-up combining patient education, individual consultations, and systematic use of PROMs. [25] Early feasibility testing has shown high acceptability, and a large multicentre evaluation is underway to assess effectiveness and cost-effectiveness. [26,27] Such structured follow-up models may help reduce uncertainty, support rehabilitation, and enable tailored support according to age, comorbidities, and disease experience, directly addressing many of the needs identified in our study.

### Strengths and limitations

To our knowledge, this is the first qualitative study to include a broad range of patients that represents the heterogenous group of PE patients. Trustworthiness of a qualitative study can be described by the three terms; credibility, dependability and transferability. [24,28–30] In this study, credibility has been increased by including a heterogenous group of participants with a variety if experiences across age, gender, cancer status and pregnancy status. By doing so, the research question can be assessed from different aspects, thus adding to a richer variation of what is studied. [28] Another way of increasing credibility, which was also done in this study, was that the analysis was performed by three researchers, who

all reached agreement of the analysed data. [28] Dependability can be increased by using a semi-structured interview guide, which was done in this study. In this way, the participants where asked the same questions at every interview, thus securing consistency in the questions asked. Transferability, although up to the reader to assess this, has been sought increased by describing thoroughly how the study has been performed, along with a thorough description of the results substantiated by deliberate quotes. [28]

Some limitations exist; there is the possibility of selection bias, meaning patients that choose to participate in research projects tend to have more symptoms and more emotional distress, meaning that the patient population might not be completely representative of the general PE population. These findings should also be viewed in the context of the Danish healthcare system, where follow-up care is publicly funded but not standardized for PE, which may have shaped participants' perceptions of unmet needs and expectations for rehabilitation. Another limitation was that one of the intended focus group interviews ended up only having one participant. This group was supposed to have included more women with pregnancy-related PE, but as women experiencing this are scarce, it was not possible to find substitutes. While the single-participant format deviates from standard focus group methodology and may have reduced data richness, we chose to include the case because the themes that emerged were consistent with those from other interviews. More broadly, the relatively small sample size (n = 17) and the limited number of participants with cancer (n = 2) or pregnancy-related PE (n = 1) restrict transferability of the findings.

## Conclusion and implications

This study showed that the patients' experiences, worries and needs are different according to where they are in their lives, which indicates that interventions aimed at improving outcomes for these patients should be targeted accordingly. The confusion about follow-up also shows that there is room for improvement regarding both communication between patients and healthcare professionals, but also for proper rehabilitation programmes. Larger studies are needed to identify and plan these properly.

## Supporting information

**S1 Table. Interview guide.**
(DOCX)

**S2 Table. COREQ checklist.**
(PDF)

## Author contributions

**Conceptualization:** Caroline Sindet-Pedersen, Nina Nouhravesh, Simone Hofman Rosenkranz, Morten Lamberts, Thomas Kümler, Gunnar Gislason, Nina Føns Johnsen, Anette Arbjerg Højen.

**Data curation:** Caroline Sindet-Pedersen, Sophie Fredslund Madsen, Anette Arbjerg Højen.

**Formal analysis:** Caroline Sindet-Pedersen, Nina Nouhravesh, Simone Hofman Rosenkranz, Gunnar Gislason, Nina Føns Johnsen, Anette Arbjerg Højen.

**Funding acquisition:** Caroline Sindet-Pedersen.

**Investigation:** Caroline Sindet-Pedersen, Nina Nouhravesh, Sophie Fredslund Madsen, Morten Lamberts, Thomas Kümler, Gunnar Gislason, Nina Føns Johnsen.

**Methodology:** Caroline Sindet-Pedersen, Nina Nouhravesh, Simone Hofman Rosenkranz, Morten Lamberts, Nina Føns Johnsen, Anette Arbjerg Højen.

**Project administration:** Caroline Sindet-Pedersen, Gunnar Gislason, Anette Arbjerg Højen.

 

**Resources:** Sophie Fredslund Madsen, Thomas Kümler.

**Software:** Caroline Sindet-Pedersen, Nina Nouhravesh, Anette Arbjerg Højen.

**Supervision:** Simone Hofman Rosenkranz, Thomas Kümler, Gunnar Gislason, Nina Føns Johnsen, Anette Arbjerg Højen.

**Writing – original draft:** Caroline Sindet-Pedersen.

**Writing – review & editing:** Caroline Sindet-Pedersen, Nina Nouhravesh, Simone Hofman Rosenkranz, Sophie Fredslund Madsen, Morten Lamberts, Thomas Kümler, Gunnar Gislason, Nina Føns Johnsen, Anette Arbjerg Højen.

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
