## [Decision Letter · Decision Letter 0]

16 Aug 2025

Dear Dr. Sindet-Pedersen,

Thank you for submitting your manuscript to PLOS ONE. After careful consideration, we feel that it has merit but does not fully meet PLOS ONE’s publication criteria as it currently stands. Therefore, we invite you to submit a revised version of the manuscript that addresses the points raised during the review process.

We look forward to receiving your revised manuscript.

Kind regards,

Yoshihiro Fukumoto

Academic Editor

PLOS ONE

Journal Requirements:

This study has been funded by the Danish Heart Foundation, Grant number: A9530.

And by an unrestricted grant from Læge Sofus Carl Emil Friis og Hustru Olga Doris Friis' Legat

This study has been funded by the Danish Heart Foundation, Grant number: A9530. And by an unrestricted grant from Læge Sofus Carl Emil Friis og Hustru Olga Doris Friis' Legat

This study has been funded by the Danish Heart Foundation, Grant number: A9530.

And by an unrestricted grant from Læge Sofus Carl Emil Friis og Hustru Olga Doris Friis' Legat

7. Please amend your authorship list in your manuscript file to include author Nina Johnsen , , Anette Højen ..

8. Please amend the manuscript submission data (via Edit Submission) to include author Nina Føns, Anette Arbjerg Højen.

Reviewers' comments:

Reviewer's Responses to Questions

**Comments to the Author**

1. Is the manuscript technically sound, and do the data support the conclusions?

Reviewer #1: Partly

Reviewer #2: Yes

2. Has the statistical analysis been performed appropriately and rigorously?

Reviewer #1: N/A

Reviewer #2: Yes

3. Have the authors made all data underlying the findings in their manuscript fully available?

Reviewer #1: No

Reviewer #2: Yes

4. Is the manuscript presented in an intelligible fashion and written in standard English?

Reviewer #1: No

Reviewer #2: Yes

Reviewer #1: Thank you for giving me the opportunity to review your manuscript entitled “The Physical and Psychological Well-Being After a Pulmonary Embolism Across Age and Comorbidities – Evidence from Focus Group Interviews.” I very much enjoyed reading your manuscript.

Summary:

The manuscript "The Physical and Psychological Well-Being After a Pulmonary Embolism Across Age and Comorbidities – Evidence from Focus Group Interviews" explores the long-term physical and psychological effects experienced by patients who have had a pulmonary embolism (PE). The study employs a qualitative approach using focus group interviews with patients recruited from an outpatient thrombosis clinic. The findings highlight that PE survivors frequently experience persistent symptoms such as fatigue and dyspnea, with varying levels of emotional and physical impact based on age, comorbidities, and risk factors such as cancer or pregnancy-related PE. Younger patients often report greater disruptions in their daily lives and a heightened sense of vulnerability, whereas older patients tend to attribute symptoms to aging. The study also identifies a "medication conundrum", where patients view anticoagulation therapy as lifesaving yet simultaneously feel a sense of chronic illness due to long-term medication use. Additionally, the study underscores a significant gap in follow-up care, with patients expressing frustration over the lack of post-PE guidance, rehabilitation strategies, and reassurance regarding recovery. The authors conclude that a personalized approach to post-PE care, tailored to age and comorbidity profiles, is necessary to improve patient outcomes. While the study provides valuable insights, further exploration of structured rehabilitation programs and targeted interventions for different patient groups could enhance its clinical applicability.

Major Comments:

1. Theory Contribution:

- The study presents important patient-centered insights into post-PE recovery, but the theoretical framework could be more explicitly defined. The authors identify a gap in post-PE care and propose a more individualized, multidisciplinary approach; however, the manuscript would benefit from a clearer articulation of how these findings advance existing theories on chronic illness management, rehabilitation, and patient-centered care. The concept of the "medication conundrum" is particularly compelling and could be further expanded—does this align with known models of chronic disease burden, or does it present a novel perspective? Discussing how these findings contribute to new patient support frameworks would enhance the study’s theoretical impact.

2. Methods:

The qualitative approach using focus group interviews is appropriate for exploring patient experiences, but additional details on study design, data collection, and analysis are needed for transparency and reproducibility. The manuscript should clarify:

- What was the justification for the sample size?

- How did the authors determine that saturation was reached?

3. Results:

The study presents valuable insights into post-PE experiences, but the results section could be better structured to highlight key themes more clearly. While the findings are significant, their impact could be strengthened by:

- Summarizing main themes in a table format for better readability (e.g., theme, representative quotes, implications).

- Providing a more detailed comparison between younger and older patients’ experiences. Are there any quantifiable differences in reported symptom burden?

Minor Comments:

1. In-text citations: You may ensure consistent formatting throughout the manuscript.

2. Study settings and participants: You may revise lines 13–19 for correct punctuation to improve clarity.

3. Line numbers: You may ensure continuous numbering throughout the document rather than restarting on each new page or section.

4. Patient quotations: You may consider presenting them in a table format to improve readability and flow instead of embedding them extensively within the text.

5. Tables:

- You may ensure that text in the first row is written horizontally, not vertically.

- You may confirm that numbers in the second column are formatted horizontally, not vertically.

Reviewer #2: This study provides valuable qualitative insights into the lived experiences of PE patients, particularly the interplay between age, comorbidities, and post-PE well-being. The findings have significant potential to inform patient-centered care strategies and highlight critical gaps in post-PE rehabilitation.

Major Concerns and Recommendations

1. Sample Size and Composition

- Comment: Small sample size (n=17), particularly for subgroups (e.g., 2 cancer patients, 1 pregnancy-related PE), limits transferability. The single participant focus group deviates from the standard focus group methodology, potentially reducing data richness.

- Recommendation: Acknowledge these limitations more explicitly in the discussion. Despite being singular, consider adding a statement on how the pregnancy-related PE case aligns with broader themes to justify its inclusion.

2. Methodological Transparency

- Comment: Limited details on how researchers resolved coding discrepancies.

- Recommendation: Clarify the process of resolving disagreements during coding (e.g., consensus meetings, third-party arbitration) to enhance dependability.

3. Ethical and Reproducibility Considerations

- Comment: Data availability is restricted due to GDPR, which is understandable but limits reproducibility.

- Recommendation: Provide a more detailed anonymized raw data summary (e.g., de-identified quotes or themes) in supplementary materials to improve transparency.

5. Conflict of Interest (COI)

- Comment: While COIs are declared, the potential influence of pharmaceutical funding (e.g., Bayer, Pfizer) on interpretation is not addressed.

- Recommendation: Add a brief statement affirming that funders had no role in study design, data interpretation, or manuscript preparation.

Minor Revisions

1. Abstract: Clarify the “unrestricted grant” source in the funding statement.

2. Results:

- Define abbreviations at first mention (e.g., “OAC” in Table 1).

- Standardize participant identifiers (e.g., “Ma (38)” vs. “K (79)”—ensure consistency in pseudonyms).

3. Discussion:

- Expand on how cultural context (Danish healthcare system) may influence follow-up care perceptions.

- Compare findings to similar qualitative studies (e.g., Danielsbacka et al., 2021) to contextualize originality.

Recommendation: Accept with Minor Revisions

**Do you want your identity to be public for this peer review?** For information about this choice, including consent withdrawal, please see our For information about this choice, including consent withdrawal, please see our Privacy Policy .

Reviewer #1: **Yes:** Ibrahim SalehIbrahim Saleh

Reviewer #2: **Yes:** Noha YasenNoha Yasen

While revising your submission, please upload your figure files to the Preflight Analysis and Conversion Engine (PACE) digital diagnostic tool, https://pacev2.apexcovantage.com/ . PACE helps ensure that figures meet PLOS requirements. To use PACE, you must first register as a user. Registration is free. Then, login and navigate to the UPLOAD tab, where you will find detailed instructions on how to use the tool. If you encounter any issues or have any questions when using PACE, please email PLOS at . PACE helps ensure that figures meet PLOS requirements. To use PACE, you must first register as a user. Registration is free. Then, login and navigate to the UPLOAD tab, where you will find detailed instructions on how to use the tool. If you encounter any issues or have any questions when using PACE, please email PLOS at figures@plos.org . Please note that Supporting Information files do not need this step.. Please note that Supporting Information files do not need this step.

---

## [Author Response · Author response to Decision Letter 1]

17 Oct 2025

The response to reviewers can be found in the cover letter, it is not possible for me to put them in here, as i have also revised a table.

---

## [Decision Letter · Decision Letter 1]

18 Nov 2025

Dear Dr. Sindet-Pedersen,

Thank you for submitting your manuscript to PLOS ONE. After careful consideration, we feel that it has merit but does not fully meet PLOS ONE’s publication criteria as it currently stands. Therefore, we invite you to submit a revised version of the manuscript that addresses the points raised during the review process.

We look forward to receiving your revised manuscript.

Kind regards,

Yoshihiro Fukumoto

Academic Editor

PLOS ONE

Journal Requirements:

Reviewers' comments:

Reviewer's Responses to Questions

**Comments to the Author**

Reviewer #1: All comments have been addressed

Reviewer #2: All comments have been addressed

2. Is the manuscript technically sound, and do the data support the conclusions?

Reviewer #1: Yes

Reviewer #2: Partly

3. Has the statistical analysis been performed appropriately and rigorously?

Reviewer #1: Yes

Reviewer #2: N/A

4. Have the authors made all data underlying the findings in their manuscript fully available?

Reviewer #1: Yes

Reviewer #2: Yes

5. Is the manuscript presented in an intelligible fashion and written in standard English?

Reviewer #1: Yes

Reviewer #2: Yes

Reviewer #1: Thank you for letting me review your revised manuscript. The authors have adequately addressed all the reviewer comments. I have no further comments to add.

Reviewer #2: This manuscript presents a relevant and well-motivated qualitative analysis of physical and psychological well-being following pulmonary embolism (PE). The topic is important and underrepresented in the literature, and the authors successfully highlight patient perspectives across different ages and comorbidity backgrounds. The findings are valuable and have potential clinical implications for post-PE follow-up and rehabilitation.

However, several aspects require revision:

• Provide clearer justification for the sampling strategy and explain more precisely how data saturation was determined.

• Expand on researcher reflexivity and positionality, as these aspects are essential for evaluating qualitative rigor.

• Clarify coding procedures (inductive vs theory-informed), describe how agreement was achieved, and specify whether member checking was conducted.

• Introduce theoretical frameworks (Leventhal’s model, biographical disruption) earlier in the manuscript instead of only in the Discussion.

• Strengthen the ethics section with a clearer explanation of the waiver for qualitative interviews.

• Improve language consistency and correct minor grammatical errors.

Overall, I recommend Major Revision, as the study has strong potential and will significantly benefit from improved methodological transparency and clearer integration with existing theoretical models.

**Do you want your identity to be public for this peer review?** For information about this choice, including consent withdrawal, please see our For information about this choice, including consent withdrawal, please see our Privacy Policy .

Reviewer #1: **Yes:** Ibrahim SalehIbrahim Saleh

Reviewer #2: **Yes:** Noha samy yasenNoha samy yasen

---

## [Author Response · Author response to Decision Letter 2]

19 Dec 2025

Point by point response

Reviewer #2: This manuscript presents a relevant and well-motivated qualitative analysis of physical and psychological well-being following pulmonary embolism (PE). The topic is important and underrepresented in the literature, and the authors successfully highlight patient perspectives across different ages and comorbidity backgrounds. The findings are valuable and have potential clinical implications for post-PE follow-up and rehabilitation.

Author response:

We sincerely thank the reviewer for the thoughtful and positive feedback. We are pleased that the reviewer recognizes the relevance of the topic, the value of highlighting patient perspectives across age and comorbidity groups, and the potential clinical implications of the findings.

However, several aspects require revision:

1. Provide clearer justification for the sampling strategy and explain more precisely how data saturation was determined.

Author response:

Thank you for this comment. We have revised the Methods section to clarify the sampling strategy and the assessment of data saturation. We now explicitly describe the use of purposeful maximum variation sampling to capture a broad range of experiences across age, sex, comorbidities, and PE-related risk factors, and explain how focus groups were composed to facilitate in-depth discussion through shared characteristics. We also clarify that meaning saturation was assessed iteratively, with recruitment continuing until two consecutive focus groups yielded no new relevant codes or perspectives, indicating that saturation had been reached.

Mehtods, study setting and participants, page 4, line 121

Participants were recruited using a purposeful maximum variation sampling strategy to capture a broad range of experiences with PE. Sampling criteria included age, sex, comorbidities, and PE-related risk factors (including pregnancy, infection, cancer, thrombophilia, and recurrence). To facilitate meaningful interaction and in-depth discussion, focus groups were strategically composed based on shared characteristics, such as age group, pregnancy-related PE, or cancer status, thereby ensuring a common frame of reference beyond having experienced PE. This approach allowed exploration of both shared and contrasting experiences across patient groups. After each focus group, the research team discussed emerging findings and compared them with insights from previous interviews. Recruitment continued until two consecutive focus groups yielded no new relevant insights, indicating that saturation had been reached

2. Expand on researcher reflexivity and positionality, as these aspects are essential for evaluating qualitative rigor.

Author response:

Thank you for this important comment. We have expanded the Methods section to explicitly address researcher reflexivity and positionality. We now describe the interdisciplinary composition of the research team (clinicians, nurses, and health researchers), and how reflexivity was addressed through ongoing team-based discussions and critical reflection during data collection and analysis.

Mehtods, study setting and participants, page 5, line 143

The research team comprised clinicians, nurses, and health researchers with experience in thrombosis care and qualitative research. Researchers involved in data collection and analysis had prior clinical and academic experience with PE, which may have influenced the research process. Reflexivity was addressed through ongoing interdisciplinary discussions of emerging findings, consideration of alternative interpretations, and critical reflection on how researchers’ professional backgrounds and preunderstandings could shape data collection and analysis.

3. Clarify coding procedures (inductive vs theory-informed), describe how agreement was achieved, and specify whether member checking was conducted.

Author response:

Thank you for this comment. We have revised the Data analysis section to clarify that coding was conducted using inductive, data-driven thematic analysis, with theoretical frameworks applied during the interpretive phase rather than guiding initial coding. We now describe how coding was undertaken by multiple researchers and how discrepancies were resolved through iterative team discussions until consensus was reached. We have also explicitly stated that member checking was not conducted and clarified the alternative strategies used to enhance credibility.

Methods, data analyses, page 5, line 151

All interviews were audio-recorded and transcribed verbatim. Data were analysed using inductive thematic analysis, following the approach described by Braun and Clarke. This method allows for the identification of patterned meanings within the data while remaining close to participants’ expressed experiences.[14, 15] The analysis proceeded through the following phases….. … Initial coding was conducted inductively and data-driven, allowing themes to emerge without the use of a predefined coding framework. Codes were primarily semantic, focusing on explicit meanings in the data[15]. ……… Coding was undertaken by multiple members of the research team (CS, NM, SR, and AH). Analytic meetings were held regularly to discuss coding decisions, compare interpretations, and refine the developing codebook. Discrepancies were resolved through discussion until consensus was reached, thereby enhancing analytic rigor. Throughout the analytic process, new codes were added when relevant, allowing the researchers to remain open to unexpected insights. Member checking was not conducted. Instead, credibility was enhanced through investigator triangulation, iterative team discussions, and careful grounding of themes in the data….

4. Introduce theoretical frameworks (Leventhal’s model, biographical disruption) earlier in the manuscript instead of only in the Discussion.

Author response:

Thank you for this suggestion. We have revised the manuscript to introduce the relevant theoretical frameworks earlier in the Methods section, prior to the presentation of the Results. Specifically, we now describe Leventhal’s Common-Sense Model of Illness Perceptions and Bury’s concept of chronic illness as a biographical disruption (along with Wagner’s Chronic Care Model) and clarify how these frameworks informed the interpretation of the qualitative data. This ensures that the theoretical perspective is established before the Discussion

Methods, theoretical framework, page 6, line 202

To support interpretation of patients’ experiences, this study draws on established theoretical frameworks.[10-12] Leventhal’s Common-Sense Model of Illness Perceptions provides insight into how symptom experiences, fears of recurrence, and coping responses may shape life after PE.[10] Bury’s concept of chronic illness as a biographical disruption offers a lens for understanding how unexpected illness can affect identity and everyday life, particularly among younger patients.[11] Finally, Wagner’s Chronic Care Model informs the exploration of follow-up care and rehabilitation needs from a health systems perspective.[12]

5. Strengthen the ethics section with a clearer explanation of the waiver for qualitative interviews.

Author response:

Thank you for this comment. We have revised the Ethics section to more clearly explain the basis for the waiver of ethical approval, specifying that under Danish legislation qualitative interview studies that do not involve interventions or biological material do not require formal ethical committee approval. We have also clarified the informed consent procedures and participants’ rights.

Methods, Ethics, page 7, line 218

The study was approved by the data-responsible institution, the Capital Region of Denmark (approval number P-2021-684). The Danish Regional Committee on Health Research Ethics waived the requirement for formal ethical approval (reference number F-25012553), as the study did not involve biological material, clinical interventions, or changes to standard care, and qualitative interview studies are not considered biomedical research requiring ethical approval under Danish legislation. All participants received oral and written information about the study aims and procedures. Oral informed consent was obtained at inclusion, followed by written informed consent prior to participation in the focus group interviews. Participants were informed that participation was voluntary and that they could withdraw their consent at any time without consequences. All data were handled in accordance with data protection regulations

6. Improve language consistency and correct minor grammatical errors.

Author response:

Thank you for this comment. We have carefully revised the manuscript to improve language consistency, clarity, and flow, and corrected minor grammatical and stylistic errors throughout the text

---

## [Editor Report · Decision Letter 2]

9 Mar 2026

The physical and psychological well-being after a pulmonary embolism across age and comorbidities – Evidence from focus group interviews

PONE-D-24-56228R2

Dear Dr. Sindet-Pedersen,

We’re pleased to inform you that your manuscript has been judged scientifically suitable for publication and will be formally accepted for publication once it meets all outstanding technical requirements.

Kind regards,

Yoshihiro Fukumoto

Academic Editor

PLOS One
---

## [Editor Report · Acceptance letter]

PONE-D-24-56228R2

PLOS One

Dear Dr. Sindet-Pedersen,

I'm pleased to inform you that your manuscript has been deemed suitable for publication in PLOS One. Congratulations! Your manuscript is now being handed over to our production team.

Kind regards,

on behalf of

Dr. Yoshihiro Fukumoto

Academic Editor

PLOS One